# Bioactive Extracts from *Salicornia ramosissima* J. Woods Biorefinery as a Source of Ingredients for High-Value Industries

**DOI:** 10.3390/plants12061251

**Published:** 2023-03-09

**Authors:** Laura Sini Sofia Hulkko, Rui Miranda Rocha, Riccardo Trentin, Malthe Fredsgaard, Tanmay Chaturvedi, Luísa Custódio, Mette Hedegaard Thomsen

**Affiliations:** 1AAU Energy, Aalborg University, Niels Bohrs Vej 8, 6700 Esbjerg, Denmark; 2RIASEARCH, Lda., Cais da Ribeira de Pardelhas 21, 3870-168 Murtosa, Portugal; 3Department of Biology, University of Padova, Via U. Bassi 58/B, 35131 Padova, Italy; 4Centre of Marine Sciences, University of Algarve, Campus of Gambelas, 8005-139 Faro, Portugal

**Keywords:** halophytes, biorefinery, antioxidants, enzyme inhibition, pigments, phytochemicals, biomedicines, nutraceuticals, sustainability

## Abstract

Salt-tolerant plants, also known as halophytes, could provide a novel source of feedstock for biorefineries. After harvesting fresh shoots for food, the lignified fraction of *Salicornia ramosissima* J. Woods could be used to produce bioactive botanical extracts for high-value industries such as nutraceuticals, cosmetics, and biopharmaceuticals. The residual fraction after extraction can be further used for bioenergy or lignocellulose-derived platform chemicals. This work analysed *S. ramosissima* from different sources and growth stages. After pre-processing and extractions, the obtained fractions were analysed for their contents of fatty acids, pigments, and total phenolics. Extracts were also evaluated for their in vitro antioxidant properties and inhibitory effect towards enzymes related to diabetes, hyperpigmentation, obesity, and neurogenerative diseases. The ethanol extract from the fibre residue and the water extract from completely lignified plants showed the highest concentration of phenolic compounds along with the highest antioxidant potential and enzyme-inhibitory properties. Hence, they should be further explored in the context of biorefinery.

## 1. Introduction

Halophytes are salt-tolerant plants that thrive in saline environments and can be found in different locations, including seashores, marshes, and saline deserts. Several species have commercial uses in different areas, including food and cosmetics, and can be commercially cultivated in saline systems. Soil and water salinisation has increased the rate of agricultural land degradation worldwide, and therefore, the cultivation of halophytes is one of the key implementations to solve these issues as they could be used for bioremediation and valorisation of these marginal lands [1]. *Salicornia ramosissima* J. Woods (Amaranthaceae), commonly known as sea asparagus or glasswort, is an edible annual succulent halophyte present in saltmarshes from the Arctic to the Mediterranean region. The importance of *S. ramosissima* as a commercial vegetable is increasing due to its organoleptic properties, including crunchy texture, slightly salty taste, and nutritional and functional properties. It is considered a reliable substitute for salt (NaCl) and, therefore, a promising functional ingredient to prevent cardiovascular diseases with appropriate levels of protein, dietary fibre, and minerals [2,3,4,5].

*S. ramosissima* can be cultivated in salt-affected marginal lands or hydroponic and aquaponic systems using saline water, including seawater [6,7,8]. However, as the plant matures, it lignifies, making it unpalatable. Due to the high salt content accumulated in the plant matrix, it is appropriate for animal fodder only when blended with other feedstuffs [9,10]. Therefore, this woody residual fraction is often considered agricultural waste; however, it could be utilised as a feedstock for biorefinery to bring additional value to farmers and provide a way to maximise material valorisation. In this regard, two concepts can be applied, depending on the plant’s growth stage: green biorefinery from partly lignified plants or more traditional lignocellulose biorefinery from completely lignified plants after seed production. The green biorefinery approach, where the biomass is first fractionated to green juice and fibre residue, often targeting the production of biochemicals, feed products, and biofuels, has been previously tested for *Salicornia sinus-persica*, *Salicornia bigelovii, Salicornia dolichostachya,* and *Salicornia europaea* [11,12,13,14].

In multi-product biorefinery, the value-added compounds with high market value can be produced to improve the process’s overall feasibility before residual fractions are utilised for bioenergy. Due to adaptation to extreme environmental conditions and a high natural defence against predators and pests, *Salicornia* spp. produce high concentrations of bioactive secondary metabolites, such as phenolic acids, flavanols, flavones, and flavanones [3,15,16]. These metabolites have reported health-beneficial properties, including antioxidant, anti-inflammatory, and anti-diabetic activities [7,15,17,18,19]. Intake of these compounds can prevent the onset of different diseases, such as cancer, hypertension, and cardiovascular diseases [3,4,20]. Moreover, studies suggest that *S. ramosissima* extracts exhibit photoprotective effects against UV radiation [4,21] and protective effects against testicular toxicity [22].

Botanical extracts or bioactive compounds with different purities could be used in different commercial areas, including nutraceuticals, pharmaceuticals, and cosmetics [16,23,24]. These are high-value industries, and the market size of nutraceuticals is forecasted to reach USD 650.5 billion by 2030. The leading force behind this growth is the increased demand for dietary supplements and functional foods [25]. The natural skin care market is forecasted to reach USD 11.87 billion by 2030, and the demand for natural cosmetics has increased due to changes in consumer behaviour [26]. The interest in biopharmaceuticals is also driven by the trend of shifting from synthetic medicine to plant-derived drugs [27].

However, the existing literature considering the bioactivity and nutritional characteristics of *S. ramosissima* has been mostly focused on fresh food-grade plants [4,18,22,28]. The number of studies considering fully mature or fractionated *Salicornia* biomass for its exploitation for bioactive compounds is limited, and only a few studies have analysed the non-food waste fraction [29] or the effect of the growth stage on the concentration of bioactive compounds. One study showed that the content of flavonoids in *S. herbacea* increases as the plant matures [30]. After extraction and production of value-added compounds, residual fractions could be utilised to produce lignocellulose-derived biochemicals or bioenergy. Considering this, biogas and bioethanol production from *Salicornia* spp. fibres and juice have been previously assessed [11,31].

In this study, the green juice fractions and botanical extracts from partly and completely lignified *S. ramosissima* plants were analysed for their total contents of fatty acids, chlorophylls, carotenoids, and phenolic compounds. In vitro antioxidant activity was also evaluated using assays with different antioxidant mechanisms. The in vitro enzyme inhibition activity was measured against enzymes related to dementia and neurogenerative diseases (such as Alzheimer’s and Parkinson’s), diabetes, obesity, and skin issues (such as hyperpigmentation and acne). Based on the obtained results, possible applications for the extracts and potential of *S. ramosissima* waste fractions for halophyte-based biorefinery are discussed.

## 2. Materials and Methods

### 2.1. Raw Material

Fresh, non-food grade, partly lignified *S. ramosissima* biomass was obtained from two different producers: Les Douceurs du Marais in France (FR) and Riasearch in Portugal (PT). *S. ramosissima* is native to these regions and is already produced (e.g., Les Douceurs du Marais) or collected from the wild for commercial purposes, and the locations have the potential for expansion of biosaline agriculture; thus, the potential for integrated halophyte-based biorefineries. In Les Douceurs du Marais, the plants were produced on an organic open-field farm in the marsh exposed to tidal changes on the west coast of France near La Turballe. Partly lignified shoots were harvested at the vegetative stage right after the food production period and the start of lignification in May 2020. The data from the closest weather station show that during the last month of cultivation, the average daily maximum and minimum temperatures were 18.6 °C and 9.8 °C, respectively, and the extreme maximum and minimum temperatures were 28.7 °C and 4.9 °C, respectively, and the total rainfall was 35.9 mm divided into 7 days with precipitation [32]. The plants obtain their water uptake from seawater (approximately 3.5 dS/m), but the salinity may vary depending on the occurrence of precipitation or droughts.

In Riasearch, biomass was cultivated in a sandy soil bed in a greenhouse. During the growth phase, plants were irrigated two times a day with a mixture of brackish water aquaculture effluent, which also fertilised the plants, and freshwater to keep the soil salinity at approximately 1.2 dS/m. Additional light fixtures were not used, and during the months with high temperatures and UV radiation, partial removable shade structures were used to protect the plants from drying and premature lignification. On the coast of Central Region of Portugal, during the summer months, the lowest average daily minimum temperature is 15.1 °C (June), and the highest average daily maximum is 24.4 °C (August) [33]. The plants were germinated at the end of February 2020, and for partly lignified biomass, aerial parts were harvested after 26 weeks of cultivation.

Dry, completely lignified, and woody *S. ramosissima* was also obtained from Riasearch. Plants were cultivated under the same conditions as those harvested at the earlier growth stage, but they were allowed to produce seeds before harvesting approximately 8 months after germination. After harvesting, the lignified plants were air-dried for approximately 5 weeks in mesh trays in shade before shipping. The plants were carefully handled after drying to avoid seed loss. The biomass batches and considered fractions and extracts are summarised in Figure 1.

Fresh biomass batches were first fractionated into green juice and fibre residue fractions using a screw press. The green juice fractions were centrifuged at 4000 rpm for 20 min, filtered using a Whatman filter (GE Healthcare), and freeze-dried using a ModulyoD freeze dryer (Thermo Scientific, Waltham, MA, USA). The fibre residue was dried overnight at a 60 °C fan oven, homogenised using a knife-miller to achieve a particle size of less than 2 mm, and stored at room temperature (RT).

The completely lignified plant material was first rinsed with water, dried at RT, and size-reduced into pieces of less than 2 cm using an agricultural straw shredder (AM55, J. N. Jensen og Sønner, Agerskov, Denmark). The dried, shredded biomass was stored at RT. The dry matter (DM) and ash concentration of the fractions were determined according to the protocols by the National Renewable Energy Laboratory (NREL) [34,35].

### 2.2. Extract Preparation

From the fibre fraction of partly lignified *S. ramosissima*, after screw press, botanical extracts were prepared using water, 70% aqueous ethanol (EtOH), and *n*-hexane as solvents, and the traditional Soxhlet apparatus with 100 mL extraction chambers. The sample size was approximately 10 g for water and EtOH extractions and 5 g for *n*-hexane extraction. All extractions were run as parallel experiments. The extraction time was 8 h for water and 6 h for organic solvents. Excess solvent was removed using a rotary evaporator, and EtOH and water extracts were freeze-dried. As the fibre residue from partly lignified plants from Portugal was not available for study, only the biomass from France was used in the extractions.

Completely lignified biomass was extracted with a pilot-scale Soxhlet using 25 L of demineralised water. The amount of biomass used was 2 kg with a dry matter content of 88.5%, and the extraction time was 8 h. It must be noted that the pilot-scale equipment does not have the same particle retention as the lab-scale Soxhlet, and in order to avoid the smallest particles in the extract phase, the shredded biomass was sifted through a 2 mm sieve. Only particles with a size more than 2 mm were used for the extraction. The obtained water extract was spray-dried using an inlet temperature of 130 °C and outlet temperature of 80 °C. All dried extracts were re-solubilised in the corresponding solvents at a final concentration of 10 mg/mL and used in the assays.

### 2.3. Chemical Characterisation of Samples

#### 2.3.1. Determination of Fatty Acids

The fatty acid (FA) profile was determined from the *n*-hexane extract of fresh *S. ramosissima* fibre residue. In transesterification, approximately 0.15 g of lipid extract was dissolved in 1 mL of 0.5 M sodium hydroxide in methanol (MeOH) at 90 °C. Samples were cooled to RT, 1 mL of boron trifluoride and 0.5 mL of hydroquinone solution were added, and samples were kept at 90 °C for another 5 min. For phase separation, 4 mL of saturated NaCl water solution and 3 mL of n-heptane were added. The non-polar fraction was recovered and analysed using gas chromatography (Clarus 500, Perkin Elmer, Waltham, MA, USA) with a capillary column (Elite-WAX, 30 m × 0.25 mm ID × 0.25 μm, Perkin Elmer). Helium was used as a carrier gas, and the temperature program was set to 1 min at 150 °C, heating 10 °C/min until reaching 240 °C, and 10 min at 240 °C. Mass spectrometry was used for the detection and quantification of fatty acids.

#### 2.3.2. Determination of Total Phenolic Compounds

The total phenolic contents (TPC) of the extracts, at the concentration of 10 mg/mL, were determined using the Folin-Ciocalteau assay described by Velioglu et al. [36] and adapted to 96-well plates. The plates were incubated for 90 min at RT, protected from light, and the absorbance was read at 650 nm using a microplate reader (EZ Read 400, Biochrom, Cambridge, United Kingdom). Results were expressed as the amount of gallic acid equivalents (GAE) in the dried extract using a calibration curve (R^2^ = 0.997).

The total flavonoid content (TFC) was also estimated in the dried extract at a concentration of 10 mg/mL, using the method developed by Pirbalouti [37] adapted to 96-well plates. Aliquots of 50 µL of the samples were mixed with 50 µL of 2% aluminium chloride in MeOH solution. The plates were incubated for 10 min at RT and read at 405 nm. Results were expressed as the amount of quercetin equivalent (QE) per gram of dried extract using a calibration curve (R^2^ = 0.999).

Total condensed tannins (TCT) of dried extracts at 10 mg/ ml were determined using a method described by Li et al. [38] using p-dimethylaminocinnamaldehyde hydrochloric acid (DMACA-HCl). Briefly, 10 μL of extracts were mixed with 200 μL of 1% DMACA in MeOH and 100 μL of 37% HCl. Plates were incubated for 15 min at RT and read at 640 nm. Results were expressed as the amount of catechin equivalent (CE) using a calibration curve (R^2^ = 0.991).

Total anthocyanidins (TAC) were measured as cyanin chloride equivalent (CCE) in the dried extract based on a calibration curve (R^2^ = 0.991). The method developed by Mazza et al. [39] was followed and adapted to 96-well microplates. In brief, aliquots of 20 µL of the samples at 10 mg/mL were mixed with 20 µL 95% EtOH containing 0.1% HCl and 160 µL 1 M HCl. The absorbance was read at 492 nm.

#### 2.3.3. Determination of Photosynthetic Pigments

The total concentrations of chlorophylls (CHL) and carotenoids (TCA) were determined from green juice fractions and EtOH extracts, as described by Lichtenthaler and Wellburn [40]. The absorbance was measured at 470 nm, 649 nm, and 665 nm using a UV-visible spectrophotometer (Genesys 50, Thermo Scientific, Waltham, MA, USA). The following equations [40] were used to calculate the concentration of pigments:CHL *a* = 13.95 × A_665_ − 6.88 × A_649_(1)
CHL *b* = 24.96 × A_649_ − 7.32 × A_665_(2)
TCA = (1000 × A_470_ −2.05 × CHL *a* − 114.8 × CHL *b*)/245(3)

### 2.4. In Vitro Antioxidant Activity Assays

Antioxidant properties were tested in vitro using radical-based and metal-based assays. *S. ramosissima* extracts were first tested at a concentration of 10 mg/mL, and the absorbances were read using a microplate reader. Antioxidant activities were calculated as a percentage relative to a control sample. For the samples with activities more than 50%, a minimum of eight different concentrations were evaluated to calculate the half-maximal effective concentration (EC_50_).

#### 2.4.1. Radical-Based Antioxidant Activity

Radical scavenging activity was tested against 2,2-diphenyl-1-picrylhydrazyl (DPPH), 2,2′-azinobis-(3-ethylbenzothiazoline-6-sulfonic acid) (ABTS), and nitric oxide (NO). For all assays, 1 mg/mL gallic acid was used as the positive control.

The DPPH assay was carried out using the method developed by Brand-Williams et al. [41] adapted to 96-well microplates by Moreno et al. [42]. A sample of 22 µL was mixed with 200 µL of 120 µM DPPH in EtOH solution. Samples were incubated for 30 min in the dark, and the absorbance was read at 492 nm.

A protocol described by Re et al. [43] was used to determine the ABTS radical scavenging activity. The 7.4 mM ABTS solution was prepared by mixing 100 mL of ABTS with 100 mL of 2.6 mM potassium persulphate in the dark and RT and incubating overnight. The final ABTS solution was diluted with EtOH to obtain an absorbance of approximately 0.7 at 734 nm. In the ABTS assay, 10 µL of the extract was mixed with 190 µL of the final ABTS solution, and plates were incubated for 6 min in the dark and read at 650 nm.

In the NO scavenging assay developed by Baliga et al. [44], aliquots of 50 µL of sample solution and 10 mM sodium nitroprusside were mixed, and plates were incubated for 90 min at RT. Afterwards, 50 µL of Griess reagent (Sigma-Aldrich, Lisbon, Portugal) was added, and the absorbance was read at 562 nm.

#### 2.4.2. Metal-Based Antioxidant Activity

In metal-based assays, iron chelating activity (ICA), copper chelating activity (CCA), and ferric reducing antioxidant power (FRAP) were tested. Positive control samples for metal-based assays were 1 mg/mL gallic acid for FRAP and 1 mg/mL ethylenediaminetetraacetic acid (EDTA) for chelating activity. All metal-based assays were carried out following Megías et al. [45], with minor modifications.

In the ICA assay, 30 µL of sample solution was mixed with 200 µL of distilled H_2_O and 30 µL of 0.01% aqueous FeCl_2_, the plates were incubated for 30 min, 12.5 µL of 40 mM aqueous ferrozine was added, the plates were further incubated for 10 min, and absorbance was read at 562 nm.

For the CCA assay, a 30 µL sample was mixed with 200 µL of 50 mM sodium acetate buffer, 100 µL of 0.005% aqueous CuSO_4_, and 6 µL of 4 mM aqueous pyrocatechol violet, and plates were immediately read at 620 nm.

The FRAP assay was performed by mixing 50 µL of the sample with 50 µL of distilled H_2_O and 50 µL 1% potassium ferrocyanide. The plates were incubated for 20 min at 50 °C oven, 50 µL of 10% aqueous trichloroacetic acid and 10 µL of 0.1% aqueous FeCl_3_ were added, the plates were incubated for another 10 min at RT, and read at 650 nm.

### 2.5. In Vitro Enzyme Inhibition Assays

Enzyme inhibition activity of 10 mg/mL sample solutions was analysed in vitro using spectrophotometric methods adapted to 96-well plates. Drugs already on the market were used as a reference: acarbose (10 mg/mL, anti-diabetic drug), arbutin (1 mg/mL, tyrosinase inhibitor), galantamine (1 mg/mL, dementia treatment), and orlistat (1 mg/mL, drug used to support weight-loss). Results were expressed as the percentage of inhibition.

The assay used for α-amylase inhibition was developed by Xiao et al. [46] based on the reaction between iodine solution and starch. Aliquots (40 µL) of sample, 0.1% boiled potato starch suspension and 100 U/mL α-amylase in 0.1 M sodium phosphate buffer solution (pH 6.9) were mixed, plates were incubated at 37 °C for 10 min, 20 µL of 1 M HCl and 100 µL of iodine solution (5 mM I_2_ and 5 mM KI in distilled H_2_O) were added, and absorbance was read at 570 nm. Results were calculated using two negative control samples, one without an enzyme (blank, 100% inhibition) for calculations and one with an enzyme for colour correction:α-Amylase inhibitory activity [%] = (A_570_ sample − A_570_ colour control)/A_570_ blank × 100%(4)

Inhibitory activity against α-glucosidase *Saccharomyces cerevisiae* was determined as described by Custódio et al. [47]. An aliquot of 50 µL of sample solution was mixed with 100 µL of 1 U/mL α-glucosidase in phosphate buffer, and plates were incubated at 25 °C for 10 min. Afterwards, 50 µL of 5 mM p-nitrophenyl-α-d-glucopyranoside was added, plates were incubated for another 5 min at 25 °C, and absorbance was read at 405 nm.

Tyrosinase inhibition activity assay was performed as described by Trentin et al. [48] by mixing 70 µL of sample solution with 30 µL of 333 U/mL fungal tyrosinase solution, incubating plates for 5 min in RT, adding 110 µL of substrate solution (2 mM L-tyrosine diluted in 25 mM potassium phosphate buffer, pH 6.5), and incubating plates for 45 min at RT before reading the absorbance at 405 nm.

Inhibition of acetylcholinesterase (AChE) and butyrylcholinesterase (BuChE) were analysed using a method developed by Ellman et al. [49]. A sample of 20 µL was mixed with 140 µL 0.02 M sodium phosphate buffer (pH 8.0) and 20 µL of 0.28 U/mL enzyme solution. Plates were incubated for 15 min at 25 °C, and 10 µL of acetylcholine iodide or butyrylcholine iodide substrate in 4 mg/mL buffer solution, and 20 µL of 5,5′-dithiobis-(2-nitrobenzoic acid) (Ellman’s reagent) in 1.2 mg/mL EtOH solution was added. Plates were incubated for another 15 min at 25 °C and read at 405 nm.

The protocol used by McDougall et al. [50] was used to measure the porcine pancreatic lipase inhibition activity. A sample of 20 µL was mixed with 200 µL 100 mM Tris-HCl buffer (pH 8.2), 20 µL of 1 mg/mL enzyme solution, and 20 µL of substrate solution (5.1 mM 4-nitrophenyl dodecanoate in EtOH). Plates were incubated for 10 min at 37 °C, and absorbance was read at 405 nm.

### 2.6. Statistical Methods

Results are given as mean values with standard deviation marked in brackets. For the extraction yields, FA, phenolic acids, HCA, and pigments, the samples were tested in triplicate (*n* = 3), while for phenolic compounds, antioxidant activity and enzyme inhibition activity, the samples were tested in sexuplicate (*n* = 6). One-way analysis of variance (ANOVA) combined with the Tukey honest significant difference (HSD) test was used to evaluate significant differences between the results, and significantly different results are denoted with different letters. EC50 values were determined for antioxidant activities using an online tool by AAT Bioquest Inc. [51].

## 3. Results and Discussion

### 3.1. Extraction Yields

Extraction yields and fractions considered in the study are presented in Table 1. The unfiltered juice fraction corresponded to more than 80% of the fresh weight of partly lignified *S. ramosissima* from France and 66.7% of biomass from Portugal. The biomass from Portugal was fractionated using a lab-scale single-auger juicer, whereas French biomass was juiced using a pilot-scale double-auger juicer with higher fractionation performance. The green juice fraction had a high ash content due to water-soluble salts accumulated in succulent plant tissue. The green juice obtained from the biomass from France exhibited especially high ash content, 81.8%, which could be explained by plants’ exposure to seawater flooding and more succulent texture compared to the more woody phenotype from Portugal. The results are aligned to those previously reported for other *Salicornia* spp., as Alassali et al. [11] found 61.1% ash content of *S. sinus-persica* juice, and Christiansen et al. [12] reported more than 80% of the total ash from the fresh *S. bigelovii* was recovered in the juice fraction after screw press. Changes in cultivation salinity do not only affect the ash content of plants but may also change the metabolism of sugars and lipids, which has been previously shown by Hulkko et al. [13] and Duan et al. [52] for *S. europaea* and Magni et al. [53] for *Salicornia perennis*. The DM from lignified plants had a lower ash content than fibre residue from partly lignified plants, but rinsing with water could have removed some of the salts from the biomass surface. The ash content of lignified *S. ramosissima* is also greatly lower than the 43.8% ash content previously reported for *S. bigelovii* [54].

### 3.2. Fatty Acid Profile

The content of non-polar compounds in *S. ramosissima* fibre residue was found to be low (1.13%). However, the lipid profile can be an important factor when biomass is considered for nutraceutical or feed application. The FA profile of the fibre residue of *S. ramosissima* is presented in Table 2. The total detected FA consisted of polyunsaturated fatty acids (PUFA, 58.2%), followed by saturated fatty acids (SFA, 41.0%), and monounsaturated fatty acids (MUFA, 1.3%). The predominant FA was linoleic acid (34.5%), followed by palmitic acid (30.9%). The obtained ω-6 and ω-3 FA ratio of 1.5 is interesting, as a ratio lower than 5 has been reported to contribute more to the anti-inflammatory state of PUFA, thereby reducing the risk of cardiovascular diseases, cancer, and autoimmune diseases [18].

Our results are consistent with those reported by Maciel et al. [55], who found the total SFA, MUFA and PUFA in chloroform extract from fresh *S. ramosissima* shoots to be 32.44%, 6.24%, and 61.32%, respectively. However, they reported a lower ratio of ω-6 and ω-3 FA (0.51), and the major difference was the amount of detected α-linolenic acid (39.6%). Isca et al. [18] determined the whole lipophilic profile for *n*-hexane extract from vegetative *S. ramosissima* and found 31.27% SFA and only 3.29% of unsaturated FA in total. Barreira et al. [56] found that 60.8% of the total FA in *S. ramosissima* was SFA, and the contents of arachidic acid (8.6%), behenic acid (10.0%), and lignoceric acid (7.1%) were especially higher compared to results obtained in this study. The differences in the lipids profile may be explained by the different phenotypes and growth stages of *S. ramosissima*. However, no studies have been found to show the variations in the FA profile regarding the growth stage or place of origin of this species. Cultivation conditions also affect the FA profile, and additional irrigation has been shown to increase the content of MUFA and PUFA in *S. ramosissima* [18]. The biomass drying process has been shown to affect the FA composition of *S. ramosissima*, as freeze-dried samples exhibited a higher amount of PUFA compared to oven-dried samples [4].

### 3.3. Bioactive Compounds

The total amounts of compounds from different phenolic groups and the concentrations of pigments in the fractions are presented in Table 3. The EtOH extract from fibre residue and water extract from lignified plant material were the richest in terms of TPC, with contents of 41.06 mg GAE/g DM and 30.10 mg GAE/g DM, respectively. These results are higher than those previously reported for *S. ramosissima* by Lima et al. [28] for the acetone extract from 200 mM salinity-cultivated fresh shoots (12.9 mg GAE/g DM) and Silva et al. [3] for the water extract from wild-harvested plants (15.02 mg GAE/g DM). However, the study by Lopes et al. [57] reports that 74.1 mg GAE/g DM of TPC were found in the *S. ramosissima* acetone extract from wild-harvested plants. Obtained results are also higher than water and EtOH extracts from *S. europaea* [58]. No significant differences were observed in the content of TFC in extracts where TFC were detected. The concentration of TFC in extracts was lower than previously reported for *Salicornia* spp. extracted with EtOH or MeOH using conventional solid-liquid extraction, such as maceration [59]. TCT was also not detected in the study by Lopes et al. [57]. However, Lima et al. [28] reported relatively high concentrations of TCT (32.5 mg CE/g DM). Polyphenols found in *Salicornia* spp. contain a variety of compounds, such as phenolic and hydroxycinnamic acids and flavonoids, linked to several biological activities of the extracts, such as anti-inflammatory and antimicrobial effects [3,4,60]. Antioxidant and cardiovascular-protective properties have been linked to compounds previously found in *S. ramosissima*, such as derivates of the flavonoids quercetin, kaempferol, and rutin, and derivates of phenolic acids, such as chlorogenic acid, p-coumaric acid, and protocatechuic acid [60].

Many processing parameters can affect the concentration of bioactive secondary metabolites, such as the used extraction method and solvent used [3], as well as drying and storage conditions [4,61]. In addition, intra-specific variability, different plant growth stages [30], biomass fractions, and plants’ exposure to abiotic and biotic stresses also affect biomass composition and phytochemical concentration. For example, condensed tannins, also called proanthocyanidins, and other flavonoids have been associated with a protective effect on plants exposed to abiotic stresses such as intensive UV radiation, drought and cold temperatures [62]. High temperatures and waterlogging (flooded conditions) have also been reported to increase the amount of bioactive compounds, as exposure to these conditions includes the production of free radicals [15,53]. Significant differences in chemical composition can be observed between plants harvested even within the same region [53]. Duan et al. [52] showed *S. europaea* cultivated under increased salinity to be enriched in phenolic acids and flavonoids in roots and aerial parts and exhibiting upregulation in bioactive compounds, including protocatechuic acid, quercetin and kaempferol derivatives, p-coumaric acid, and ferulic acid. Cultivated plants have also been shown to have lower phenolics content than wild populations, likely due to more controlled cultivation [63]. These aspects must be taken into consideration when planning potential biorefinery processes to ensure they are robust enough to withstand variations.

Despite the bright red colour of the juice from French plants, anthocyanins were not detected in the sample. The red colour was only observed in the juice from French plants, which could be due to their cultivation methods, could have produced some protective pigments in response to abiotic stresses. However, the concentration of the pigments may have been below the limit of detection, or the stability of the compounds has been compromised due to the neutral pH of the solvent [64]. Among sugar-free anthocyanidins, pelargonidin is known to have colours ranging from orange to red, whereas cyanidin has a strong magenta colour [62]. Thus, there may be a difference in the maximum absorbance of the compounds present in the sample and cyanin chloride used as a standard.

Photosynthetic pigments were detected in the EtOH extract due to the more non-polar nature of these compounds. In the EtOH extract, the concentration of CHL *a* was higher compared to the juice fraction, with a CHL *a*/CHL *b* ratio of 2.3, whereas in the juice fractions from the French and Portuguese biomass, the corresponding ratios were 0.72 and 0.53, respectively. A similar ratio in *Salicornia* EtOH extract has been reported previously for *S. neei* [65]. The amount of total CHL in the EtOH extract was lower than that previously reported for *S. ramosissima* EtOH extracts by Barreira et al. [56] (21560 μg/g DM) but higher compared to *S. brachiata* (746.5 µg/g DM) and *S. neei* (233.3 µg/g DM) studied by Parida et al. [66] and De Sousa et al. [65], respectively. Salt stress may decrease the amount of CHL in the plants [65], which may explain the lower CHL content in the juice from French plants compared to Portuguese plants, as the plants cultivated in France were exposed to higher salinity.

The concentration of total carotenoids in *Salicornia* spp. varies between different studies [16], and all obtained results lay within the range of reported results. Carotenoids are non-enzymatic antioxidants produced in response to different stressors [67]. The different results reported for the same species are strongly dependent on the conditions where the plants have grown, and light, temperature, and salinity variations are the main factors that may lead to carotenoid production. Carotenoids have been previously reported to have a key role in the salt-tolerance mechanisms of Amaranthaceae, and cultivation salinity has been shown to impact the pigment content of *Salicornia* spp. [65,68,69].

An efficient water-based extraction process with high phenolics yield would be desirable in a biorefinery targeting the production of bioactive compounds, as using solvents such as EtOH increases the capital expenditures and operational costs of the larger-scale facility. Biomass cultivation in a controlled environment, such as a hydroponic system, could provide a possibility to modify the growth conditions and environmental stresses to enhance the production of target compounds, such as phenolic acids and flavonoids.

### 3.4. Antioxidant Activity

Salinity and other environmental stresses trigger oxidative reactions and the generation of reactive oxygen species, causing cellular damage and metabolic disorders in plants [7,70]. For halophytes, these stresses are more pronounced, and they have developed efficient antioxidant defence mechanisms to cope with extreme environmental stresses. Halophytes produce different classes of antioxidant compounds, such as phenols and carotenoids, which may have a synergic effect as radical scavengers. As seen in Figure 2, all fractions presented antioxidant activity in radical-based assays, and the water extract from the fibre residue exhibited the highest activity in a 10 mg/mL concentration. Compared to the antioxidant activity previously reported for *S. ramosissima* EtOH extract by Barreira et al. [56], the obtained extracts exhibited lower DPPH radical scavenging activity (IC50 5.69 mg/mL). The results for aqueous extracts are also higher than those reported by Faria et al. [71], who found the DPPH and ABTS radical scavenging activity of 10 mg/mL *S. neei* 80% EtOH extract to be 37.1% and 46.1%, respectively. Considering the water extract from the fibre residue, the EC_50_ value of ABTS radical scavenging activity was very similar to that reported by Lima et al. [28] for *S. ramosissima* acetone extract. The activity in the NO assay indicates that the extracts from *S. ramosissima* could have anti-inflammatory properties, as exposure to NO radicals has been directly linked to chronic inflammation [44]. Thus, the extracts could be a potential source of ingredients for dermo-cosmetics and biopharmaceuticals. Extracts from *S. europaea* and *S. brachiata* have been previously reported to have anti-inflammatory properties [72]. Chronic inflammation is involved in various diseases, including but not limited to cardiovascular diseases, diabetes, autoimmune and neurogenerative conditions, and chronic kidney disease [73]. There are no available results for the NO scavenging activity of EtOH extract due to the precipitation of the sample.

The antioxidant activity was generally more pronounced in metal-based assays (Figure 3). Considering the FRAP assay, even though the differences between the results for water and EtOH extracts are non-significant, the EC_50_ values for water extracts are much lower. Therefore, the extracts can be considered more potent, indicating the presence of strong, water-soluble antioxidant compounds in the *S. ramosissima* biomass. Unfortunately, there are no results for CCA of the green juice from Portuguese biomass due to issues with the assay. The EC_50_ values of fibre residue water extract for FRAP, ICA, and CCA were also greatly lower than those reported by Barreira et al. [56] for *S. ramosissima* EtOH extract. For the fibre residue water extract, the obtained EC_50_ (4.91 mg/mL) of CCA is very similar to the one reported by Lima et al. [28] for *S. ramosissima* acetone extract. Achieving the same antioxidant activity with water extraction and reducing the use of toxic and flammable organic solvents is highly desirable considering large-scale industrial applications. Similarly, obtaining extracts from *S. ramosissima* residues with bioactivity comparable to extracts from food-grade plants highlights the biorefinery and valorisation potential of these residues.

Overall, the green juice fractions exhibited the lowest bioactivity, whereas water and EtOH extracts from the fibre residue of French plants had the highest activity. Even though more mature plants have been shown to have higher concentrations of certain compounds with antioxidant activity [30], the extract from completely lignified Portuguese plants exhibited the highest activity only in ICA and FRAP assays. Therefore, the cultivation conditions could explain the higher antioxidant activity of French *S. ramosissima*. Plants grown in an open field have most likely been exposed to more abiotic stresses, such as high UV radiation and temperature difference, as well as higher cultivation salinity, leading to increased antioxidant production. On the contrary, *S. ramosissima* was grown in a more controlled greenhouse environment. The difference can also be observed in the juice fraction, as the juice obtained from French plants had higher antioxidant activity in all assays, except ABTS scavenging activity, compared to Portuguese phenotypes.

The mixture of different bioactive compounds present in the specific biomass fraction or extract could also contribute differently to each antioxidant mechanism, affecting the observed activities. The high concentration of antioxidants produced by halophytes has made them interesting for functional food applications and improving the nutritional quality of everyday products such as bread or pasta [2,7,58].

### 3.5. Enzyme Inhibitory Properties

The enzyme inhibitory properties of green juices and extracts are presented in Table 4. Samples displayed different inhibitory activities against the different enzymes, and the EtOH extract with 10 mg/mL concentration exhibited moderate to high inhibition activity against enzymes related to diabetes (α-glucosidase), obesity and acne (lipase), hyperpigmentation (tyrosinase), and neurogenerative diseases (AChE).

One treatment for diabetes mellitus is to limit glucose absorption using inhibitors against enzymes responsible for breaking the carbohydrates, such as α-glucosidase and α-amylase, and the potential of plant-derived extracts as inhibition agents has been investigated [74,75]. Some compounds linked to anti-diabetic properties are phenolics, flavonoids, and anthocyanidins [74]. The α-glucosidase inhibition activity of the EtOH extract (68.63%) is high, especially when compared to acarbose with the same concentration (85.61%) of 10 mg/mL. Raw water extracts, which still include a high amount of salt, had moderate α-glucosidase inhibition activity (34.12% and 8.93% for extracts from fibre residue and completely lignified biomass, respectively), which could be improved by extract purification and increasing the concentration of phenolics. There was no significant difference in the α-glucosidase inhibition activity of water extracts from partly lignified and completely lignified biomass. Considering other *Salicornia* spp., flavonoids isorhamnetin 3-*O*-glucoside and quercetin 3-*O*-glucoside isolated from Korean *S. herbacea* (syn. *S. europaea*) EtOH extract have previously shown potential for blood sugar regulation by α-glucosidase inhibition [76]. Similarly, hydroxycinnamic acid trans-ferulic acid, also found in *S. ramosissima*, has also been reported to have anti-diabetic properties [60]. Hwang et al. [77] also reported α-glucosidase inhibition of 31.9% using desalted 70% aqueous EtOH extracts from fresh shoots of *S. herbacea* with a concentration of 0.5 mg/mL.

For the EtOH extract from *S. ramosissima* fibre residue, the α-amylase inhibition activity was observed visually by the change in sample colour. However, due to precipitation in the sample, it was not possible to reliably measure the absorbance. Green juice fractions and water extracts showed only a very low inhibition of α-amylase.

Obesity, resulting from excessive accumulation of fats, affects metabolic health and its increased prevalence has led to various public health concerns worldwide [78]. To address this issue, phytochemicals with lipase inhibitory properties have been studied as a means of reducing lipid absorption [78]. *S. herbacea* has been shown to have lipase inhibition activity in vitro, and its consumption reduced plasma triglyceride and cholesterol levels in an animal model [77,79]. In addition to their use as anti-obesity agents, lipase inhibitory phytochemicals have been investigated for their potential in treating acne, a common chronic disease, as bacteria related to acne produce lipase to break down triglycerides in sebum to free FA, which then causes skin inflammation [80]. EtOH extract from *S. ramosissima* fibres showed moderate (41.74%) lipase inhibition at a concentration of 10 mg/mL. There was significant differences in the lipase inhibition activity of water extracts from fibre residue and completely lignified biomass. Together with potential anti-diabetic properties, the obtained results indicate that extracts from *S. ramosissima* could provide potential ingredients for nutraceutical and pharmaceutical applications targeting obesity and related lifestyle diseases. However, further investigation and testing are required to confirm the therapeutic properties.

Tyrosinase inhibitors are investigated as skin-whitening agents for cosmetics and for treating hyperpigmentation, but they also have an important role in food and agricultural applications for preventing products oxidation [81]. In order to preserve the appearance and nutritional properties of fresh foodstuff such as cut fruits, safer natural anti-browning agents have been investigated to replace commonly used sulfiting agents [82]. *S. ramosissima* EtOH extract (10 mg/mL) exhibited high tyrosinase inhibition activity of 71.85%. Anti-tyrosinase activity of *Salicornia* spp. has been explored in several studies. According to Ahn et al. [83], the inhibition activity of more than 50% was reached at a concentration of 60 mg/mL of *S. bigelovii* ethyl acetate extract, which is a much higher required concentration. However, the EtOH extract of *S. europaea* has shown 21.04% activity with a concentration of 1 mg/mL [84]. Sung et al. [67] also found that even low concentrations (0.1 μg/mL) of water extract from *S. herbacea* inhibit tyrosinase (>50%) and significantly reduce melanin synthesis in melanoma B16 cells. Copper is a cofactor of tyrosinase, and tyrosinase inhibition activity may be linked to the CCA of extracts [85]. However, no tyrosinase inhibition activity was observed in *S. ramosissima* fibre residue water extract, which exhibited the highest CCA in antioxidant assays. Considering dermo-cosmetic applications, *S. ramosissima* water extract supplemented cream has already been shown to reduce mechanically evoked itching (hyperkinesis), a condition related to atopic dermatitis, and regulate the skin barrier [85]. Additionally, a cream supplemented with a 3-methoxy-3-methyl-1-butanol extract from Japanese *S. europaea* has shown promising results for UVB-protective properties by improving the skin texture in areas exposed to the sun [21]. Therefore, tyrosinase inhibition activity, together with these effects, could indicate the potential for *Salicornia* extracts for use in dermo-cosmetics. However, further investigation is needed to evaluate the therapeutic properties and effects of the long-term use of extracts.

Cholinesterase enzymes are responsible for the breaking down of acetylcholine and other choline esters, which function as neurotransmitters. Targeting these enzymes with inhibitory agents and increasing the levels of neurotransmitters has been seen as a potential symptomatic treatment for neurogenerative diseases [86,87]. The EtOH extract from fibre residue (10 mg/mL) exhibited high AChE inhibition activity (68.42%), and green juice fractions showed low to moderate AChE and BuChE inhibition activities. Higher inhibition of cholinesterase enzymes was observed using green juice from Portuguese plants compared to French plants. Differences could be explained by phenotypic variations, such as different phytochemicals present in the fractions. However, a detailed metabolomic analysis would be needed to determine the different compounds present in the juice fractions. The studied water extracts did not exhibit cholinesterase inhibition activity. However, Pinto et al. [29] previously determined the AChE inhibition activity of 1 mg/mL water extract from lignified *S. ramosissima* to be 32.34% using a commercial assay kit and showed the extract to be rich in caffeoylquinic acid derivatives. Besides intra-specific variation, the difference could be due to different extraction methods and potential compound degradation, as Pinto et al. [29] used maceration in lower temperatures and shorter resident time than in Soxhlet extraction. Karthivashan et al. [88] reported in vitro AChE inhibition activities of approx. 42% and approx. 78% for 1 mg/mL desalted EtOH extract (rich isorhamnetin and acanthoside B) and enzyme-digested wild-harvested Korean *S. europaea,* respectively, and showed significant suppression in AChE activity in the mice model. The desalted EtOH extract from *S. europaea* has also been tested on subjects complaining of memory dysfunction without dementia, but regardless of some positive results concerning the comprehension of spoken language function and Stroop test results, the study has its limitations due to the small number of subjects and short duration [89]. Phytochemicals, especially phenolics, and the potential synergistic effect of different compounds in plant extract matrices have been suggested to contribute to the neuroprotective properties of botanical extracts [29]. However, further investigation is needed to reveal the potential of botanical extracts as therapeutic agents.

Considering biorefinery, value-added products targeted to the biopharmaceutical industry and cosmetics are desirable, as they often have a relatively high market value and can also be seen as some of the key applications of *Salicornia* species [72]. Full metabolomic profiling and further analysis considering the contribution of specific compounds to different bioactivity are still open for investigation. Several mixtures of phenolic compounds have shown synergistic effects [90], and these mechanisms are still unexplored to a great extent. If the botanical extracts from halophytes could be utilised as a matrix instead of purified isolated compounds, some costly downstream processing steps could be avoided. However, the phenotypical variation due to the biomass harvest stage and cultivation conditions, which could be observed when comparing Portuguese and French biomass, may cause challenges when the extracted matrix is produced for an application requiring high consistency, such as biopharmaceuticals. Generally, biorefinery processes must be designed to withstand some degree of variation in the raw material composition.

## 4. Conclusions

Bioactive properties of different residue fractions from *S. ramosissima* biomasses were assessed. The FA profile showed a high amount of PUFA. The water extract from completely lignified biomass and EtOH extract from the fibre residue fraction had the highest concentration of phenolic compounds. Aqueous extracts exhibited high antioxidant activity comparable to extracts with organic solvents, making them interesting for industrial applications. EtOH extract from fibre residue had high and moderate inhibition activity against α-glucosidase and lipase, respectively, indicating the potential for nutraceuticals and biopharmaceutical applications targeting obesity and diabetes. Tyrosinase and lipase inhibition activities also make the extracts interesting for cosmetic applications. Since raw extracts were considered, bioactivity can be improved by purification and increasing the concentration of phytochemicals. Extracts from residual fractions obtained with non-toxic solvents exhibited bioactivities comparable to fresh *S. ramosissima*, which has increased interest as a nutrient-rich commercial vegetable and potential source of bioactive compounds. Overall, residue fractions from *S. ramosissima* could be a potential source of bioactive extracts, making it interesting to investigate possible biorefinery concepts further for maximum feedstock valorisation.

## Figures and Tables

**Figure 1 plants-12-01251-f001:**
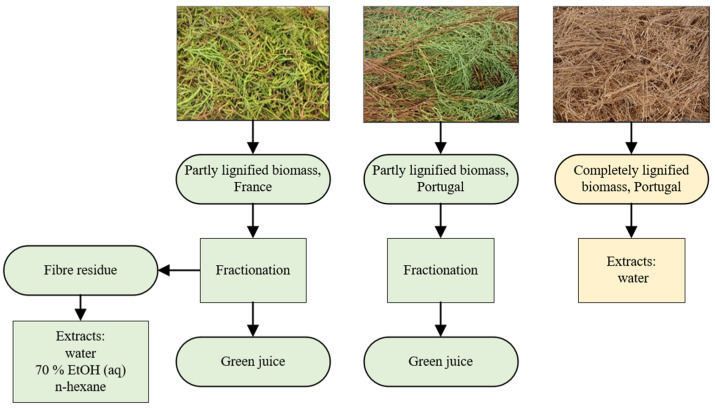
Origin of *Salicornia ramosissima* biomass batches and considered fractions and botanical extracts.

**Figure 2 plants-12-01251-f002:**
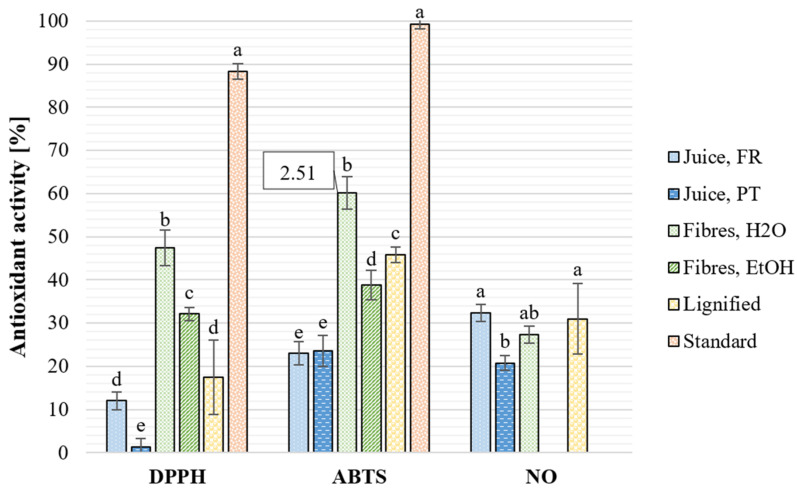
Antioxidant activity [%] of extracts from *S. ramosissima* at the concentration of 10 mg/mL in relation to blank samples with extraction solvent determined by assays with radical-based antioxidant mechanisms. DPPH: 2,2-diphenyl-1-picrylhydrazyl, ABTS: 2,2′-azinobis-(3-ethylbenzothiazoline-6-sulfonic acid), NO: nitric oxide. EC50 values [mg/mL] are presented in callout boxes for *Salicornia* samples with activity more than 50%. The standard compound is gallic acid (1 mg/mL). Different letters denote significantly different results, calculated individually for the results from each assay and for all assays (*p* < 0.001).

**Figure 3 plants-12-01251-f003:**
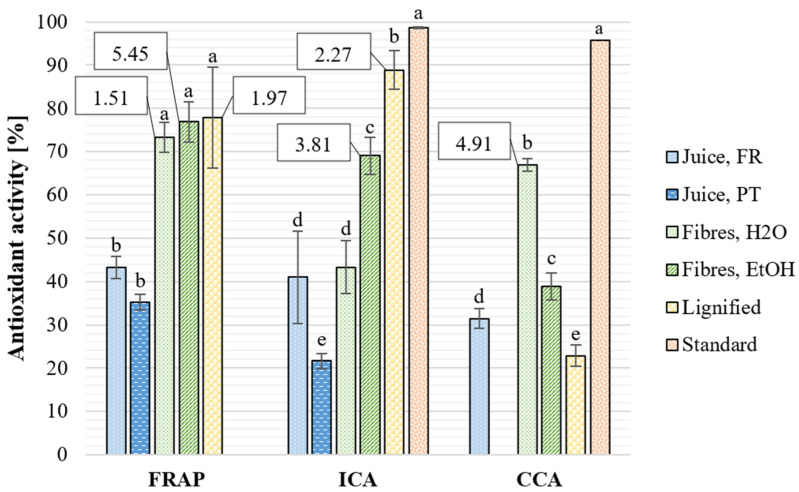
Antioxidant activity [%] of extracts from *S. ramosissima* at the concentration of 10 mg/mL in relation to blank samples with extraction solvent determined by assays with metal-based antioxidant mechanisms. FRAP: ferric reducing antioxidant power, ICA: iron chelating activity, CCA: copper chelating activity. EC50 values [mg/mL] are presented in callout boxes for *Salicornia* samples with activity more than 50%. The standard compound is ethylenediaminetetraacetic acid (1 mg/mL). Different letters denote significantly different results, calculated individually for the results from each assay and for all assays (*p* < 0.001).

**Table 1 plants-12-01251-t001:** Yields of *Salicornia ramosissima* extracts. Results are given as [*w*/*w*%] on the basis of fresh biomass weight for dry matter content and dry matter weight for extractives and ash. n/a: not available. Results are expressed as mean values with standard deviations (in brackets).

Fraction	Dry Matter [*w*/*w*%]	H_2_O Extract [*w*/*w*%]	EtOH Extract [*w*/*w*%]	*n*-Hexane Extract [*w*/*w*%]	Ash [*w*/*w*%]
Juice, FR	7.73 (0.01)	n/a	n/a	n/a	81.83 (0.39)
Juice, PT	6.47 (0.21)	n/a	n/a	n/a	64.76 (0.13)
Fibre residue, FR	25.49 (0.27)	33.68 (2.31)	31.45 (0.59)	1.13 (0.06)	20.31 (0.11)
Lignified biomass	89.65 *	18.83 *	n/a	n/a	17.28 (0.98)

* Extraction was run as one batch in a pilot-scale system; not possible to give a standard deviation.

**Table 2 plants-12-01251-t002:** Fatty acid profile of the lipids extracted with *n*-hexane from French *Salicornia ramosissima* fibre residue. Results are given as [% FA/total FA], n.d.: not detected. Results are expressed as mean values with standard deviations (in brackets).

Fatty Acid	Fibres, *n*-Hexane Extract [% FA/total FA]
Myristic acid C14:0	0.4 (0.4)
Palmitic acid C16:0	30.9 (2.3)
Palmitoleic acid C16:1	n.d.
Stearic acid C18:0	2.7 (0.1)
Oleic acid C18:1	1.3 (0.1)
Linoleic acid C18:2	34.5 (0.7)
α-Linolenic acid C18:3	23.7 (1.4)
Arachidic acid C20:0	1.0 (0.9)
Behenic acid C21:0	3.9 (0.1)
Lignoceric acid C24:0	2.1 (1.8)
ΣSFA	41.0 (0.9)
ΣMUFA	1.3 (0.1)
ΣPUFA	58.2 (2.0)
ω-6/ω-3	1.5 (0.1)

**Table 3 plants-12-01251-t003:** Bioactive compounds in *S. ramosissima* fractions. TPC: total phenolic compounds [mg GAE/g DM] (*p* < 0.001), TFC: total flavonoids [mg QE/g DM] (*p* = 0.240), TCT: total condensed tannins [mg CE/g DM], TAC: total anthocyanidins [mg CCE/g DM] (*p* < 0.001), CHL: total chlorophyll [µg/g DM] (*p* < 0.001), TCA: total carotenoids [µg/g DM] (*p* < 0.001), n.d.: not detected or concentration lower than the limit of detection, n/a: not available. Results are expressed as mean values with standard deviations (in brackets). Different letters denote significantly different results, calculated individually for each compound group.

Fraction	TPC [mg GAE/g DM]	TFC [mg QE/g DM]	TCT [mg CE/g DM]	TAC [mg CCE/g DM]	CHL (a + b) [µg/g DM]	TCA [µg/g DM]
Juice, FR	1.34 (0.54) ^d^	n.d.	n.d.	n.d.	15.42 (3.15) ^c^	73.16 (1.41) ^b^
Juice, PT	3.66 (0.74) ^d^	n.d.	n.d.	n.d.	39.26 (0.22) ^b^	52.31 (2.43) ^c^
Fibres, H_2_O	18.84 (1.04) ^c^	3.61 (0.67) ^a^	n.d.	1.46 (0.31) ^a^	n/a	n/a
Fibres, EtOH	41.06 (4.09) ^a^	3.22 (0.45) ^a^	n.d.	n/a	1446.37 (13.99) ^a^	261.55 (1.56) ^a^
Lignified	30.10 (1.97) ^b^	3.86 (0.71) ^a^	n.d.	0.53 (0.11) ^b^	n/a	n/a

**Table 4 plants-12-01251-t004:** Enzyme inhibition activity [%] of reference compounds and 10 mg/mL *S. ramosissima* stock solutions in relation to blank samples. n.d.: not detected or activity lower than the limit of detection, n/a: not available. Reference compounds were tested with 1 mg/mL, except acarbose (10 mg/mL). Results are mean values with standard deviations (in brackets). Different letters denote significantly different results, calculated individually for the results from each enzyme assay. For all assays (*p* < 0.001), except for lipase inhibition assay (*p* = 0.002).

Fraction	α-Amylase [%]	α-Glucosidase [%]	Lipase [%]	Tyrosinase [%]	AChE [%]	BuChE [%]
Juice, FR	2.10 (0.41) ^bc^	5.46 (4.98) ^c^	19.21 (9.09) ^b^	10.84 (2.66) ^b^	16.11 (1.91) ^c^	24.03 (2.77) ^b^
Juice, PT	3.29 (1.16) ^ab^	12.91 (1.73) ^c^	21.40 (8.77) ^b^	n.d.	28.44 (4.60) ^b^	46.25 (1.74) ^a^
Fibres, H_2_O	4.08 (0.72) ^a^	34.12 (2.10) ^b^	n.d.	n.d.	n.d.	n.d.
Fibres, EtOH	n/a	68.63 (8.85) ^a^	41.74 (8.93) ^a^	71.85 (5.42) ^a^	68.42 (1.59) ^a^	17.82 (1.77) ^c^
Lignified	1.10 (0.47) ^c^	38.93 (3.90) ^b^	21.13 (8.64) ^b^	n.d.	n.d.	n.d.
Acarbose	61.18 (1.30)	85.61 (0.89)	n/a	n/a	n/a	n/a
Orlistat	n/a	n/a	82.33 (1.52)	n/a	n/a	n/a
Arbutin	n/a	n/a	n/a	41.46 (1.51)	n/a	n/a
Galantamine	n/a	n/a	n/a	n/a	88.88 (0.96)	41.41 (4.11)

## Data Availability

The data generated and analysed during the study are available from the corresponding author on request.

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
