# Peer review of "Bioactive Extracts from Salicornia ramosissima J. Woods Biorefinery as a Source of Ingredients for High-Value Industries"

_plants, 2023, doi:10.3390/plants12061251_

Round 1
Reviewer 1 Report
The present manuscript has interesting information in terms of bioactive compunds extracted from different sources of Salicornia.
However, authors authors stated along the manuscript that "growth stage and cultivation conditions could explain the differences in the antioxidant activity of juice fraction obtained from French and Portuguese".
The big concern here is that indeed this is true, especially in terms of salinity growth conditions, the authors didnt emphazised the proper growth conditions in their samples.
The initial description of the sample collection is not clear and there is a lack of details regarding the conditions of salinity (mM), light, temperature, nutrients, irrigation, etc. were used for each population studied. Although it is obvious, it has also been reported in many studies that the bioactive compounds of Salicornia are influenced by these conditions. Then the authors should start from these conditions and then present the results and discuss them appropriately.
It is also not clear why these two samples from France and Portugal were chosen.
In the results it is observed that sometimes both populations are compared, but if authors are not clear about the starting point of the growth conditions, this could not be reproducible in the future, especially if it is intended to obtain bioactive compounds.
In other sections, for example, Fatty acid is indicated only for the population of France, the reason is not clear.
The graphs must be improved, some bars were left without statistical analysis.
Based on the suggested changes, the discussion of the results and conclusions should be improved.
Reviewer 2 Report
Hulkko et al., investigated the bioactive extract preparation from Salicornia ramosissima J. woods byproduct or waste as a source of value added-product for various industries.
I have conducted my review of the present paper and would like to appreciate author's conceptualization and execution of this study.
Overall MS has been written very well with scientific judgment and support for the results.
All selected and conducted experiments justify the potential use of Salicornia ramosissima J. woods byproduct or biorefinery waste.
This study will definitely add value to the market, basically, waste utilization for value-added products.
MS can be accepted and published in its present form.
Reviewer 3 Report
-The paper does not make a significant contribution to new knowledge in the discipline.
-The research idea lacks novelty.
-Authors did not identify the main bioactive compounds responsible for the studied biological activities.
Round 2
Reviewer 1 Report
I consider that the manuscript was improved substantially and I recommend for publish.
However I suggest to include more relevant and actual refrences:
Magni, N.N.; Veríssimo, A.C.S.; Silva, H.; Pinto, D.C.G.A. Metabolomic Profile of Salicornia perennis Plant’s Organs under Diverse In Situ Stress: The Ria de Aveiro Salt Marshes Case. Metabolites 2023, 13, 280. https://doi.org/10.3390/metabo13020280
Cárdenas-Pérez, S., Piernik, A., Chanona-Pérez, J.J., Grigore, M.N., Perea-Flores, M.J., 2021. An overview of the emerging trends of the Salicornia L. genus as a sustainable crop. Environ. Exp. Bot. 191. https://doi.org/10.1016/j.envexpbot.2021.104606
Duan H, Tiika RJ, Tian F, Lu Y, Zhang Q, Hu Y, Cui G, Yang H. Metabolomics analysis unveils important changes involved in the salt tolerance of Salicornia europaea. Front Plant Sci. 2023 Jan 20;13:1097076. doi: 10.3389/fpls.2022.1097076
Hulkko, L.S.S., Turcios, A., Kohnen, S. et al. Cultivation and characterisation of Salicornia europaea, Tripolium pannonicum and Crithmum maritimum biomass for green biorefinery applications. Sci Rep 12, 20507 (2022). https://doi.org/10.1038/s41598-022-24865-4
Author Response
The authors want to thank the Reviewer for their kind feedback and great suggestions for references. We are familiar with the studies, and they were previously excluded as they consider other Salicornia species rather than S. ramosissima. However, as suggested, we have now included them and cited them in the manuscript to give a better insight into the functionality of the species in the genus.
Reviewer 3 Report
According to the comments, the authors did not improve the article and additional experiments are needed.
Author Response
We are sorry to hear that our arguments regarding the novelty and contribution of the study were not sufficient for the Reviewer. In the initial review report, the Reviewer did not suggest any specific areas that needed to be changed or elaborated. We hope that the additional references added in this revision round will provide further improvement to the manuscript.